# Eight-Year Health Risks Trend Analysis of a Comprehensive Workplace Health Promotion Program

**DOI:** 10.3390/ijerph17249426

**Published:** 2020-12-16

**Authors:** Antti Äikäs, Pilvikki Absetz, Mirja Hirvensalo, Nicolaas Pronk

**Affiliations:** 1Faculty of Sport and Health Sciences, University of Jyväskylä, 40014 Jyväskylä, Finland; mirja.hirvensalo@jyu.fi; 2Faculty of Social Sciences, Tampere University, 33014 Tampere, Finland; pilvikki.absetz@gmail.com; 3HealthPartners Institute, HealthPartners, Bloomington, MN 55420, USA; Nico.P.Pronk@HealthPartners.Com

**Keywords:** workplace health promotion, health risks, effectiveness, implementation, program evaluation, risk management

## Abstract

Research has shown that workplace health promotion (WHP) efforts can positively affect employees’ health risk accumulation. However, earlier literature has provided insights of health risk changes in the short-term. This prospective longitudinal quasi-experimental study investigated trends in health risks of a comprehensive, eight-year WHP program (*n* = 523–651). Health risk data were collected from health risk assessments in 2010–2011, 2013–2014, and 2016–2017, applying both a questionnaire and biometric screenings. Health risk changes were investigated for three different time-periods, 2010–2013, 2014–2017, and 2010–2017, using descriptive analyses, *t*-tests, and the Wilcoxon Signed Rank and McNemar’s test, where appropriate. Overall health risk transitions were assessed according to low-, moderate-, and high-risk categories. Trend analyses observed 50–60% prevalence for low-, 30–35% for moderate-, and 9–11% high-risk levels across the eight years. In the overall health risk transitions of the three time-periods, 66–73% of participants stayed at the same risk level, 13–15% of participants improved, and 12–21% had deteriorated risk level across the three intervention periods. Our findings appear to indicate that the multiyear WHP program was effective in maintaining low and moderate risk levels, but fell short of reducing the total number of health risks at the population level.

## 1. Introduction

A central objective of worksite health promotion (WHP) is to prevent health risk accumulation among the working age population. Earlier, mostly cross-sectional studies indicate that poor employee health (e.g., depression, high stress, high body mass index (BMI), musculoskeletal problems, heavy use of alcohol, sedentary lifestyle) is associated with increased medical and pharmacy claim costs [1,2,3,4], reduced productivity [5,6,7,8,9], and absenteeism [7,10].

In response to cross-sectional and observational studies, WHP efforts have been implemented in order to reduce health risks among employees, and related research has been conducted that tests the effectiveness of interventions [11]. Reviews have emerged concluding how WHP appears to be effective in its objectives to improve health, and may also generate a financial return on investment [11,12,13,14]. However, there are also studies which have cast doubt on conclusions that WHP programs are effective or cost-effective [15,16,17,18]. Regardless of a scientific effectiveness debate, employers’ interest towards positive impacts on health, productivity, and return-on-investment has generated a demand for potential WHP solutions [14,19]. From a health risk management perspective, these efforts should focus on two major objectives: (1) keeping employees in low-risk categories, and (2) moving high-risk individuals into low-risk categories [20].

Several studies [7,20,21,22] have noted the importance of investigating health risk stratification. Health risks may be described as an accumulation of an increased likelihood of illness and disease, which is linked to health behaviors and environments in which individuals participate and engage over the course of their lifespan [23,24,25,26]. A person can either be apparently healthy, be at an increased risk level, or have multiple health risks and chronic conditions [7,27]. In the WHP context, risk levels are usually identified with a health risk assessment (HRA) or biometric screening, and stratification of health risks is based on the number of risks present [11,21]. For example, Edington (2001) defined those with zero to two health risks as low-risk, three to four as medium-risk, and five or more as high-risk [21]. Loeppke and colleagues used the Edington’s risk levels in a health risk cohort study of 2606 employees from 2008 to 2009, and categorized 55.7 percent of the population as low-risk, 31.1 percent as increased-risk, and 13.2 percent as high-risk [20].

Earlier WHP intervention studies offer insights into health risk reduction with mostly positive or no-change results. At the beginning of the millennium, Yen and colleagues observed that the number of people at low-risk status increased slightly from 70.1% to 71.3% during the General Motors two-year program [28]. Henke et al. (2011) reported mixed outcomes in health risks during a four-year program at Johnson & Johnson [29]. Positive changes were observed in blood pressure, cholesterol, nutrition, physical activity, and tobacco use, whereas there were negative changes in high stress and depression, and no change in obesity levels [28]. Blake et al. (2013) reported positive findings after a multilevel workplace wellness intervention in a National Health Service setting, but only reductions in sedentary lifestyle were statistically significant [22]. Byrne et al. (2011) reported statistically significant improvements in the proportion of employees at low risk, such as their physical inactivity, sufficient fruit and vegetable intake, smoking, seatbelt usage, and high stress during a seven-year, incentive-based worksite wellness program [30]. However, prevalence of high cholesterol, high blood pressure, overweight, and existing medical problems did not decrease in the study´s trend analysis [30].

To summarize the previous literature, comprehensive, multiyear programs tend to lower prevalence in several health risk categories [20,29,30,31], but the slope of such changes is relatively flat [28,29,32] and also includes negative findings [29,30,31,33]. Furthermore, the vast majority of studies report results of relatively short intervention periods from one to three years [11,20,28,31,32,33], especially when considering the context of health risk accumulation as the population ages.

From the existing literature, we also know that context, design process, execution process, and evaluation phases provide a framework for effective implementation [7,12,34]. A well-designed intervention presents a scalable and sustainable solution that provides intervention dose, target populations, and specific aims that are context-relevant [27,35]. Furthermore, well-executed intervention components include easy access coaching sessions, helping participants to learn and maintain healthy behaviors, sufficient participation levels, long-lasting engagement, and cost-effective implementation, connected with ongoing evaluation [12,34,36].

The purpose of this study was to examine long-term health risk trends in response to a comprehensive WHP program. Baseline measurements in 2010–2011 were followed up over the course of eight years according to two follow-up assessments, one in 2013–2014 and another in 2016–2017. As a result, the study complements and adds to the scientific literature from two perspectives: (1) What kind of health risk trends can be noticed among all employees across three time points? And (2), did the multi-year intervention help to reduce or mitigate health risk accumulation among the study’s male-dominated population during the eight-year period?

Noteworthily, a pragmatic, in-depth evaluation of the same program was conducted earlier in the context of scientific dissemination and implementation frameworks and noted successful integration of evidence-informed best practices, but also identified improvement potential for a clear path toward sustainability, integration with occupational health care, and support from middle-management supervisors [34,37]. In addition, the present comprehensive program was observed to achieve high penetration and implementation levels, reasonable participation rates, and a moderate health impact on the target population during the eight-year-period [38]. Similar prospectively designed, multi-component, multidimensional, and longitudinal evaluations are rare.

## 2. Materials and Methods

### 2.1. Study Design

This case study focuses on a multi-year health risk evaluation of a WHP in a single company using a prospective, longitudinal, quasi-experimental cohort study design without a control group. This study´s reporting protocol followed the STROBE guidelines (see [App app-ijerph-17-09426]). Health risk trend measurements were made for three time-points representing the periods from 2010‒2011, 2013‒2014, and 2016‒2017, respectively. Trend analyses investigated the prevalence of health risks at each of these measurement points. Three time-period analyses considered participants´ health risk changes from 2010 to 2013 (Phase 1), from 2014 to 2017 (Phase 2), and from 2010 to 2017 (total period), respectively. Similar trend and cohort analysis methods have been used by Byrne and colleagues (2011) and Loepkke and colleagues (2010) [20,30].

### 2.2. Intervention

A comprehensive intervention (ENSO) was delivered over an eight-year period from 2010 to 2017 in a Finnish wood supply company, Stora Enso Metsä. The organization´s main focus was to buy, harvest, and transport wood for Stora Enso mills throughout the nation. At the beginning, the employer had altogether over 100 business units nationwide and over 600 employees, which were the participants of the program. Subcontractors and family members were not in a target population. The ENSO program was a tailored version of a comprehensive WHP concept designed and produced by the provider of the program, 4event Ltd., Lahti, Finland. The ENSO program can be classified as a comprehensive program, including all five elements that are based on the Healthy People 2010 definition: health education, supportive environment, integration into organization´s structure, linkage to related programs, and worksite screening [39].

The primary focus of ENSO was to improve the health and well-being of every employee, and the intervention and its services have been described earlier in detail [37,38]. The ENSO program consisted of six main components: assessments of health risks, services for all employees, targeted services, local playmaker network, several communications tools, and ongoing program management. Altogether, 4event Ltd. executed 27 different services (e.g., wellness coaching, local fitness camp, targeted phone calls) from 2010 to 2013 and 46 different services (e.g., mental coaching, Sleep well camp, Body & Mind) from 2014 to 2017. The health risk assessment was a launching element of ENSO, and its purpose was to define employees’ health and their needs for targeted services. If a participant had poor health status or health risk, she/he was invited into targeted service by the service provider [37,38]. However, the ENSO program did not contain prescriptions or influence health care professionals´ medication procedures. 

Noteworthily, the intervention´s very first four years emphasized low-effort, pleasant lifestyle changes, nutrition, and physical activity, whereas the last four years concentrated more on the psychosocial aspect of workplace wellness, such as emphasizing workplace climate, stress management, and mental resources [37,38]. The midpoint shift in major content enabled us to study different implementation impacts on three time-periods, from 2010 to 2013 (Phase 1), from 2014 to 2017 (Phase 2), and from 2010 to 2017 (total period). See further information about implementation components in [App app-ijerph-17-09426]. 

Finally, it should be noted that ENSO had *voluntary* and *primary prevention* features, whereas statutory occupational health care was available for the employees via three different health providers during the program continuum. The occupational health care units were informed about results of the health risk assessment, but due to administrative reasons, they were not involved either in the design nor implementation processes until the very last years of the intervention. 

### 2.3. Participants

Each year, all employees of Stora Enso Metsä were participants of the program. Most participants were white-collar workers, that is, executives, local forest officers, organization officials, and a smaller proportion were blue-collar workers, such as lumberjacks and terminal workers in the organization´s units connected into Stora Enso’s eight paper mills in Finland. The total number of participants progressively declined during the course of the study as follows: 651, 634, 630, 625, 530, 526, 523, and 523 during the years 2010 through 2017. The annual employee turnover rates were 11.0%, 8.2%, 4.1%, 17%, 5.1%, 3.6%, 5.5%, and 7.8%, respectively. The largest drop in the total number of staff happened in 2014, when statutory labour negotiations took place and, as a result, the local forest officers and lumberjacks in Western Finland were no longer a part of the company. Due to the fact that the drop in total participant number might have had an influence on health risk trends over time, we tested for differences between baseline results among follow-up HRA participants compared to participants at baseline only ([App app-ijerph-17-09426]). The tests clearly show that employees who took part only at baseline were older (*p* < 0.000), had more health risks (*p* < 0.001), and were less likely to belong to the low health risk group (*p* < 0.020) than multi-year HRA participants. Every employee, despite whether they participated in assessments of health risks, was provided free access to the ENSO program.

### 2.4. Data Collection

Every employee was invited to participate in assessments of health risks (HRA) in 2010–2011, 2013–2014, and 2016–2017. The HRA consisted of a questionnaire and biometrical screenings, which participants completed at the same time. The biometric part of assessments was executed by 4event Ltd. with a transferable Polar BodyAge^TM^, which is a technology-aided testing system [40]. Altogether, the body age protocol consisted of five physiological factors, that is, BMI, body fat percentage, systolic blood pressure, diastolic blood pressure, and the Polar own Fitness test for VO^2^max (based on heart rate variability measurement), and four performance factors, that is, the number of crunches in 60 s, a leg endurance test, a bicep curl, and a sit-and-reach test [40,41,42,43]. The HRAs were offered close to the participants´ workplaces and multiple occasions so that attendance was available throughout the day and measurement period. 

The questionnaire included demographic items (age, gender, location, personnel group) and items related to physical activity, lifestyle change, weight management, musculoskeletal disorders, and mental resources. The questionnaire was a combination of questions from an annual national survey of Health Behaviour and Health among the Finnish Adult Population, Polar Body Age test protocol, the stages of transtheoretical model, the service provider´s own questions, and the employer’s own questions [44,45,46]. Altogether, the questionnaire had 36 questions at the baseline, 68 in the first follow-up, and 66 in the second follow-up. The average response time was 15–20 min.

The health risk information was pooled from items listed in Table 1. The final data consisted of eight variables which were included in each data collection point, as well as four additional important health risk variables (e.g., smoking) which were available only at follow-up one and follow-up two.

The actual health risk definition was based on a simple 0 or 1 taxonomy, a participant either having or not having a risk. For example, physical inactivity was asked using a four-point Likert scale (low, moderate, high, very high), and a sedentary job and minimal exercise per week portrayed the low criteria and ascribed a risk. For a vitality measurement, participants were asked to select one option from a five-point scale, starting from “I seldom feel energetic and vital” ending up to “I feel energetic and lively every day”. The lowest possible answer, “seldom”, was assigned a risk. In a similar self-reporting manner, chronic pain and stress and alcohol usage were asked using a five-point Likert scale: (1) totally disagree, (2) disagree, (3) neutral, (4) partly agree, and (5) totally agree. A high level of health risk was considered by only including “totally agree” options when indicating chronic pain and stress risk. However, alcohol usage was considered risky behaviour if a respondent selected either four or five. Comparatively, musculoskeletal disorders (MSD), medication for cardiovascular disease and regular smoking were simple yes/no statements, in which the “yes” answers determined a status of a health risk. Absences due to MSD were counted as a risk if the person reported being away from work for four or more days due to MDS during the last 12 months.

A similar simple taxonomy was used in biometrical health risk screenings. The cardiovascular fitness classification was based on the seven-point scale by Shvarts and Reibold (1990), where results from very low to low were assigned as a health risk [47]. High blood pressure was categorized as a health risk if a person’s systolic blood pressure was over 139 mmHg, or diastolic pressure was over 89 mmHg. A limit value for high body mass index (BMI) has varied in earlier studies; hence, criteria previously used in well-known studies in the WHP field that stem from the National Institutes of Health in the U.S., ≥27.8 kg/m^2^ for men and ≥27.3 kg/m^2^ for women, were deployed in this study [20,21].

### 2.5. Data Analysis

Eligible employees were selected from the health risk analysis database using two inclusion criteria. First, an employee had to have participated in both health risk assessments (the biometrical screenings and the questionnaire). Second, an employee had to have data for at least six-out-of-eight or nine-out-of-twelve risk variables, depending on the data collection period used. In case of missing data, no statistical replacement methods were used.

Data on health risk trends were analyzed at three different time-points to represent the prevalence of health risks among the population, with eight health risks at the baseline in 2010–2011, and 12 health risks in 2013–2014 and 2016–2017. Health risk trends were compared descriptively, including percent point difference, count difference, or average difference.

To investigate health risk changes during the intervention, three time-periods were utilized. These analyses included only those employees who completed two HRAs and biometrical screenings either between 2010 and 2013, 2014 and 2017, or between 2010 and 2017. Phase 1 from 2010 to 2013 represented the first half of the eight-year intervention (*n* = 359), whereas Phase 2 from 2014 to 2017 represented the latter half (*n* = 255). Finally, the total period from 2010 to 2017 represented the entire continuum (*n* = 253). The time-periods were relevant for this study, since they represented different intervention phases and were also used in our earlier study, where we investigated implementation coefficients of each of the periods and overall effectiveness of the same comprehensive program [38]. It should be noted that the participants in the periods were not independent, as 215 persons took part in all three assessments (see [App app-ijerph-17-09426]).

We investigated not only changes of single health risk variables, but also individual level improvements or reductions in the total number of health risks. With this in mind, more detailed analyses were completed for the three time-periods in order to investigate total health risk accumulation in different subgroups (sex, age, and personnel group). We also included a subgroup analysis of “successful” and “unsuccessful” lifestyle change participants, which were classified in our earlier study [38]. A participant was considered to be successful if she/he met two clear criteria. First, that the person answered YES to the question, ‘‘Have you made a lifestyle change during the program?’’, and chose at least one from a list of nine lifestyle change options. Second, the person’s Polar BodyAge^TM^ result had to have been improved based on the five physiological, and the four performance factors [38,40,43]. This unique comparison allowed us to investigate the possible impact of successful/unsuccessful lifestyle changes on the accumulation of health risks. 

Edington (2001), Loeppke (2010), and Byrne et al. (2011) defined overall risk levels as follows: low (0–2 high risks), moderate (3–4 high risks), and high (5 or more risks) [20,21]. However, we used cut-off points 0–1 for low-, 2–3 for moderate-, and 4 or more for high-risk levels. The reason for using these more stringent categories is based on our data collection approach that included only eight and 12 health risks, whereas the comparison studies used 15. It is also notable that our data collection did not contain blood tests, that is, for cholesterol or fasting blood sugar. 

We explored the overall risk levels (low, moderate, and high) in the trend and time-period analyses to summarize health risk prevalence among participants. Furthermore, in the final analyses of this study, the transitions between low, moderate, and high risk levels were described with the following categories: improved, unchanged, or deteriorated. For example, a person was classified as “deteriorated” if she/he moved from a low risk level into a moderate- or high-risk group.

Statistical analyses included descriptive statistics, a paired samples t-test, the Wilcoxon Signed-Rank and McNemar´s test, where appropriate. Significance of the results was reported at 0.05, 0.01, or 0.001 levels.

### 2.6. Ethical Issues

This study followed the ethical principles of the University of Jyväskylä and the research guidelines provided by the National Advisory Board on Research Ethics in Finland [48]. The employer and all the employees provided informed consent for this research. The protocol of the study with an informed consent form was included in the HRA remapping questionnaire both in 2013–2014 and 2016–2017 and handed out to each employee. Participation in this research, as well as giving authorization to use the earlier results of assessments of health risks, was entirely voluntary. The health risk data collected were managed by the researcher and entered into IBM SPSS 26.0 for statistical analysis.

## 3. Results

### 3.1. Health Risk Trends

Table 2 describes the demographics and health risk trends at baseline and two follow-up measurements among the study population. At baseline, ~90% percent attended the health risk assessment, with ~81% at the first follow-up, and ~69% at the second follow-up. In the first follow-up, the drop-out reflected net reduction of the total number of staff, but not anymore at the second follow-up. As an overview, the populations participating HRA data were, for the most part, equivalent between three time-points. The mean age stayed almost the same between three different time-points, although the proportions of the age group ≤35 years and 46–55 years were smaller in the follow-ups than baseline. From an occupational perspective, local forest officers were the largest personnel group each time, and lumberjacks were not a part of organization and analysis after the first follow-up.

According to the trend analysis, the average number of health risks was 1.49 at baseline, 1.41 at follow-up one, and 1.30 at follow-up two, when the calculation constituted eight health risks. As the number of measured health risks altered from the baseline´s eight to follow-ups´ twelve, the average number of health risks increased to 1.59 at follow-up one, but stayed at the same level (1.30) at follow-up two. 

The prevalence of single health risks varied from 0.2 to 8.8 percent points between baseline, follow-up one, and follow-up two. When comparing the follow-up trends against baseline, lower-risk levels were noted in physical inactivity, low cardiovascular fitness, high blood pressure, high BMI, smoking, and daily alcohol usage. In contrast, prevalence of musculoskeletal disorders (MSD), four or more illness days due to MSD, medication for cardiovascular disease, low vitality, chronic pain, and self-reported stress stayed the same or at a higher level. 

A relevant finding for this study was that the distribution of overall risk levels was relatively stable, showing only slight variations in the low-risk (~55–60%), moderate-risk (~30–35%), and high-risk (~9–11%) categories, respectively. To summarize the trend analysis, the three cross-sectional populations analyzed were quite similar from a health risk trend perspective, although slight differences were found in the age group distribution, personnel groups, and among single health risk variables.

### 3.2. Health Risk Changes

In Table 3, health risk changes are reported for the three time-periods of the study. Overall, the changes in all three periods are small, which is a parallel finding with the results of the health risk trend analysis. Unique to this analysis, there was an increase in medication for cardiovascular disease during the entire program, and significant changes were observed in all three periods (*p* < 0.05). A relatively large increase was found in self-reported musculoskeletal disorders in Phase 1 (*p* = 0.001), but the change faded during the Phase 2 and the total period. The prevalence of high BMI increased during the total period (*p* = 0.023), though statistically significant BMI changes were not documented either in Phase 1 or Phase 2. Altogether, positive health risk reductions were noted among 4/8 variables during the Phase 1, among 6/12 variables during the Phase 2, and among 4/8 variables during the total period. However, none of these changes were statistically significant. 

More detailed results for health risk accumulation in the three periods and different subgroups are presented in [App app-ijerph-17-09426]. These subgroup analyses were based on sex, age group, and personnel group comparisons. These analyses also contained our earlier study´s “successful” and “unsuccessful” lifestyle change groups and their comparisons in health risks changes [38]. According to the results, these two groups were meaningful predictors of health risk accumulations during Phase 1 and the total period.

Statistically significant risk reduction was found in Phase 1´s “successful group” from 2010 to 2013 (1.33 ± 1.36 to 1.13 ± 1.18, t = 1.984, *p* < 0.05). On the contrary and during the same time-period, risk increase was documented in the “unsuccessful group” (1.33 ± 1.24 to 1.65 ± 1.41, t = −4.278, *p* < 0.001), among men (1.39 ± 1.31 to 1.56 ± 1.35, t = −2.680, *p* < 0.01), and among local forest officers (1.26 ± 1.12 to 1.43 ± 1.36, t = −2.244, *p* < 0.05). On the other hand, statistically significant changes were rare in Phase 2 and during the total period. For instance, we did not find any significant subgroup changes in Phase 2, and the only significant change during the total period was the unsuccessful group´s increase in the number of risks (1.16 ± 1.16 to 1.54 ± 1.37, t = −4.235, *p* < 0.001).

In general, the average number of risks was higher in age groups 46–55 years and ≥56 years, and among terminal workers and lumberjacks in all three periods, and among men in both Phase 1 and 2. Furthermore, the average number of health risks was low (less than two risks) and tended to grow slightly in most subgroups during the four- and eight-year follow-up periods.

### 3.3. Health Risk Accumulation

Figure 1 depicts overall risk level changes for the three time-periods. At the starting point of each phase, the proportion of low-risk participants varied from 58.4 percent to 67.6 percent. Corresponding values for moderate- and high-risk populations were between 25.3–33.0% and 7.0–8.6%, respectively. The transitions between low-, moderate-, and high-risk populations were very similar in Phase 1 and in the total period. During these time periods, the proportion of participants who stayed at the same risk level was equal (66%), and the number of deteriorated participants (19–21%) was slightly larger than improved participants (13–15%). On the contrary, in Phase 2, 15% of participants managed to improve their overall health risk level against 12% of deterioration, while most of the participants stayed in the unchanged population with a proportion of 73 percent.

More specifically, the most common groups in the analysis clearly represented employees who stayed at the low-risk level (47.4–53.4%) or at the moderate-risk level (10.3–16.5%). If a participant shifted between risk levels, then the majority of transitions was observed either from low to moderate, or moderate to low levels across the three time-periods. The transitions from or into the high-risk population were small.

## 4. Discussion

This study investigated health risk trends and participants’ health risk transitions between low, moderate, and high risk categories during an eight-year comprehensive workplace health promotion program. A relevant aspect of this study was that the program´s implementation was split at the halfway point of the study. The first part included a focus on more traditional behavior change approaches, such as nutrition and physical activity (Phase 1), whereas the latter part focused more on the psychosocial aspect, emphasizing workplace climate, stress management, and included more support for mental health resources (Phase 2). The major content change and three different measurement time-points created a unique ground for this study.

Understanding health risks is important from a health promotion perspective, since they are the source of many non-communicable diseases, loss of healthy and productive years, and a causal factor for increased health-care expenses all over the world [7,23,25,26,27]. Hence, there are two clear aims for health promotion interventions: (1) to keep healthy employees healthy, and (2) to move high-risk individuals into lower-risk categories [7,20].

A major finding of this study was that it substantiated the first aim but failed to achieve the second aim. The overall health risk levels stayed at the same rate, meaning that the majority of the population, both in the trend and time-period analyses were in the low- and moderate-risk categories, and stayed there. Furthermore, the three time-period analyses revealed that, if there were transitions between risk levels, the most common changes took place between low and moderate groups—that is, movement from or into the high-risk population was rare.

As it comes to health risk reduction and moving the high-risk population into the moderate- or low-risk category, six important observations are highlighted based on this study. First, the prevalence of measured health risks was lower than in an average working-age population [20,29,30,44]. For example, Loeppke et al. (2010) reported ~42 percent of employees having high BMI, whereas our study´s trends varied between 33–38 percent [20]. The amount of Johnson & Johnson´s employees experiencing high stress varied annually between ~7–11%, while our population´s level was ~3–5% [29]. A multiyear European health risk trend study reported physical inactivity levels between ~18–27%, whereas our study´s levels were ~15–20% [30]. Among Finnish working-age people, 22% of males and 15% of females smoke regularly, while this study´s average varied between ~9–10% [44]. Furthermore, the similarity test ([App app-ijerph-17-09426]) showed that a healthier and younger proportion of employees took part in HRA assessments. *A good health level at baseline* is, of course, a favorable premise from a maintenance perspective, but might have contributed to a floor effect when considering the aim to reduce health risks. Furthermore, the trend analysis of single health-risk variables revealed that high blood pressure and high BMI were common (>30%), every fifth or fourth employee had musculoskeletal disorders (>20%), and physical inactivity was quite common at baseline (~20%), but all other health risks considered were quite less common (<10%). These findings help us understand why health risk assessment participants showed a relatively low number of 1.3–1.5 risks per person, and why most of the health risk transitions were observed between the low- and moderate-risk categories.

Second, even though our earlier study observed a positive moderate impact on biometrical health and fitness variables [38], the program was not able to influence health risk variables in a similar manner. For instance, the prevalence of medication for cardiovascular disease was noted to increase in all three time-periods. This trend might have been derived from prescriptions for hypertension medication, which is a widely used treatment [30,49]. If an older participant was prescribed a medication due to high blood pressure, this phenomenon was out of reach of this study´s WHP context, since the intervention did not seek influence on health care procedures. Identically, the prevalence of musculoskeletal disorders, illness days to MSD, chronic pain, and self-reported stress might have been derived from permanent individual or environmental issues, which were *inaccessible* for this study´s health promotion efforts. All mentioned variables stayed the same or slightly increased in the trend or time-series follow-ups.

The third observation relates to *aging*. It is a well-known fact that when people age, the prevalence of certain health risks tends to increase [25,26]. Additionally, middle-aged people with high risk levels suffer from high risk levels also at an older age [50,51,52]. A large Finnish health and functional capacity study (*n* = 9288) showed that the male average BMI tended to rise from 27.0 to 28.1 between the age groups of 30–39 and 60–69 years [53]. The corresponding figures for women were 25.8 and 28.2 [53]. Similarly, 40% of working-aged men and 26% of working-aged women had high blood pressure (>140/90 mmHg), and the levels rose to 57% (men) and 61% (women) after the age of 65 years [53]. In this research, the changes in the participants´ BMI and blood pressure during Phase 1, Phase 2, and total period analysis of an eight-year continuum were mostly minor, which means that the net change was minimal. In this sense, our findings show that the effects aging has on health risk accumulation can be balanced.

The fourth observation refers to the health risk accumulation in different sub-groups ([App app-ijerph-17-09426]). In Phase 1 from 2010 to 2013 and the total period from 2010 to 2017, we found statistically significant differences in men, executives, local forest officers, unsuccessful, and successful groups when using the paired samples t-test. These subgroup findings give a reason to expect that some employees did not take advantage of the program’s support or were not able to achieve permanent healthier lifestyles nor avoid an increasing number of health risks. As a future recommendation, it would be of substantial interest to identify proactive means of engagement of those individuals who belong to “*successful*” and “*unsuccessful*” lifestyle change groups at an early point of implementation in order to optimize their positive impact on health risks. These kinds of course-correction methods have been supported by several researchers and include the notion of feedback loops in program design, as exemplified by the so-called 4-S model, which was used as a design evaluation tool in our first study [35,37].

Fifth, it is possible that healthy and risky behaviors were accumulated for single individuals in the male-dominated population. The conclusion is supported by longitudinal studies, where healthy (or risky) behaviors tend to be maintained for even decades [49,50,51,52]. Another point of view of health risk accumulation is that some measured outcome variables were related. For example, physical inactivity and low cardiovascular fitness were associated. Similarly, high blood pressure, high BMI, and medication use for cardiovascular disease are likely related. These explanations may explain why participants remained at the same health risk level.

Sixth, we decided to use clearly defined *stringent* cut-off points, 0–1 risks for low, 2–3 risks for moderate, and 4 or more for the high risk levels, due to the fact that our data collection consisted only of eight and twelve health risk variables. Other benchmark studies have used fifteen [20,21]. Interestingly, our health risk trend observations for low- (~55–60%), moderate- (~30–35%), and high-risk populations (~9–11%) were at a similar level as reported by Loeppke et al. (2010) who deployed less strict limits and found positive health risk reductions [20]. In this study, however, our classification scheme may have caused a lower likelihood to see the impact of the intervention, since tighter limits also mean a need for stronger impact to move participants into lower health risk levels.

When comparing this study´s results against the relevant literature, we found earlier health risk studies differing in terms of measured variables, duration, and primary context. However, studies evaluating WHP programs offer valuable insights of possible intervention effects. Loeppke et al. (2010) reported positive health risk changes of 11/15 variables during one-year intervention [20]. Henke et al. (2011) found positive changes of 6/9 variables during a four-year continuum [29]. Gold et al. (2000) observed program participation to have had a positive impact on 9/13 health risks over two years, and Byrne et al. (2011) reported improvements in 9/16 health risks over seven years [30,33]. In this study, positive and negative changes were evenly divided during the first four years 4/8, during the last four years 6/12, and during the whole period 4/8. To summarize the earlier literature and our findings, it seems that even an intervention would manage to produce health risk reductions in single variables, negative changes are also most likely to occur, and the reachability of permanent positive changes is strongly connected to health risk definitions, cut-off points, baseline level, population characteristics, and aging.

In the final analysis, our findings from the three time-periods reported 66–73% of participants maintaining, 13–15% improving, and 12–21% deteriorating in overall risk levels. A bit surprisingly, more positive observations were found in overall health risk stratification during Phase 2, even though the time-period had the lowest overall health impact according to our earlier study [38]. This discrepancy most likely ensued from the data collection of Phase 2, which consisted of 12 health risk variables and thus gave more opportunities for improvement.

According to the Natural Flow model by Edington (2001), it is expected that 61% of the population would maintain their risk level, 23% end up in a worse level, and only 16% will improve [21], but the model does not reflect benchmark values for longer follow-ups. Earlier studies have reported more positive results, such as the research by Loeppke et al. (2010) that reported how 68% unchanged, 23% improved, and 9% worsened, as well as a two-year intervention study by Pronk (2014) that noted 66% had no change, 21% improved, and 13% deteriorated [20,54]. These observations lead us to speculate on what would have enhanced the impact of this study’s specific long-term program on the reduction of health risks.

One point of view is that the content of the program should have been focused more directly on employees who experienced the health risks targeted by the program. Instead, the main focus of the program was on themes which were most likely suitable for everyone (i.e., nutrition, physical activity, workplace culture). If done so, it might have increased the “effectiveness” of targeted services, but at the same time, it might have reduced the level of participation because the prevalence of health risks was mostly less than <30%, and usually employees with more severe health risks are the least likely ones to participate [36]. Furthermore, it should be kept in mind that our earlier findings indicated a reasonable participation rate that was sufficient to produce an impact [38].

Another perspective would have been to focus more strongly on the high-risk population. This approach is supported by well-documented observations that financial savings come from managing diseases [14,32,55,56]. Yet, this notion may be challenged by the point of view stating that it is important to target the group with the highest costs, while also offering program opportunities to those employees who are at the medium- or low-risk levels [7,57]. However, although our analysis did not investigate these matters at this point, future research should evaluate financial data to justify the decisions that support comprehensive WHP programming.

Another potential improvement opportunity would have been closer co-operation with occupational health care. Our first study noted that a closer relationship was pursued, but not achieved in 2012, and it would have been a valuable resource in managing health risks, especially in the high-risk group, since at least some of the risks measured in this study must have also been registered during health care appointments [37]. Likewise, other implementation studies have supported the aspect of co-operation in order to reach the program´s objectives [58].

This study had several limitations that should be noted. The current study lacked a control group, and the results and findings should be considered in a quasi-experimental context. The data collection consisted of eight and twelve health risk variables, whereas some benchmark studies have used as many as fifteen. In addition, the data collection did not offer blood tests, which are commonly used to identify health risks, such as high cholesterol and fasting blood glucose. Additionally, our analyses did not investigate health risks from a socioeconomical, education, or income perspective. Finally, the workforce was mostly male and white-collar. Similar results might not be observed in a more gender-balanced or a low-wage workforce.

Balanced against the noted limitations, this study also showed several major strengths. Notably, its eight-year duration, three different measurement points, and health risk assessments including both biometrical and questionnaire data provided a unique database for analysis. Furthermore, this study investigated the health risk changes in a real-world context with a long-term research effort, and complemented the earlier design and implementation analysis of the same comprehensive program [37,38].

## 5. Conclusions

In summary, this study’s results show that the accumulation of health risks as a function of aging was slowed down, but not reduced in a mostly male employee population over an eight-year period. The reasons behind succeeding in one aim and failing in another were multidimensional. Our findings indicate the difficulty to lower health risks at the organizational level, particularly when the prevalence of health risks is not high, if the health promotion efforts do not reach the nature of a risk and if there are subgroups who do not engage with the intervention. However, our findings agree with earlier literature that a comprehensive WHP program can affect employees´ health and slow health risk accumulation, as noted when compared to a situation where no intervention is provided or engagement levels are low [7,20,21,22,28,29,30,31,32,33]. Future research should investigate the reasons behind “unsuccessful” lifestyle change groups in order to increase the effectiveness of health risk prevention plans. Finally, even though this study´s intervention has been analyzed from several implementation perspectives, an economic analysis is needed to complete the overall evaluation.

## Figures and Tables

**Figure 1 ijerph-17-09426-f001:**
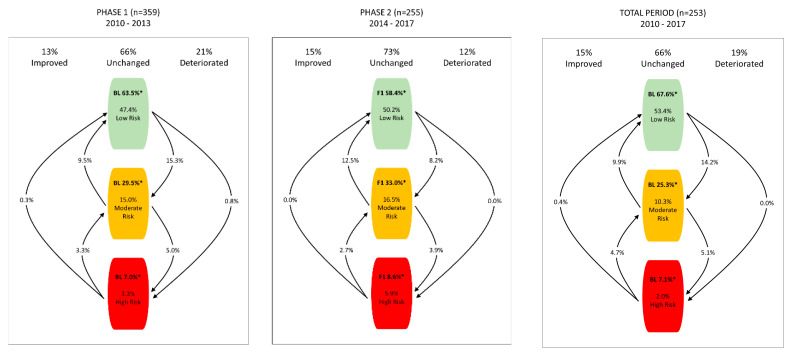
Overall Health Risk Level Transitions During Phase 1, Phase 2 and Total period; * The proportion of low-, moderate- and high-risk participants at the starting point of each time-period.

**Table 1 ijerph-17-09426-t001:** Health risks and definitions.

Health Risk	Time Points *	Measurement	Source	Definition of High Risk **
Physical inactivity	B,F1,F2	Questionnaire	self-reported	“Low” responses to a four-point (low, moderate, high, top) self-reported physical activity level during the last six months. The description of the “low” was set as follows: − “At work you sit most of the day or your work does not require much physical effort.” − “Your daily exercise is minimal (0–1 times per week) and you walk rarely.” − “You seldom sweat or become breathless during exercise.”
Low cardiovascular fitness	B,F1,F2	Biometric screening	Measured submax test VO_2max_ in ml/kg/min and categorized	1 (very Low) or 2 (low) cardiovascular fitness in 7 point scale by Shvarts and Reibold 1990 [47]
High blood pressure	B,F1,F2	Biometric screening	Measured mmHg	Systolic ≥ 140 or diastolic ≥ 90
High BMI	B,F1,F2	Biometric screening	Measured weight (kg) and height (m) and expressed as kg/m^2^	≥27.8 kg/m^2^ men, ≥27.3 kg/m^2^ women
Low vitality	B,F1,F2	Questionnaire	self-reported	“Yes” to *“I seldom feel energetic and vital.”*
Musculoskeletal disorder	B,F1,F2	Questionnaire	self-reported	“Yes” to *“Do you have musculoskeletal problems or injuries that prohibit you from exercising?”*
Illness days due to MSD	B,F1,F2	Questionnaire	self-reported	“4 or more days” to “*Have you been missed from work due to MSD illness during the past 12 months?”*
Medication for cardiovascular disease	B,F1,F2	Questionnaire	self-reported	“Yes” to *“Are you on medication for high blood pressure or for heart disease?”*
Chronic pain	F1,F2	Questionnaire	self-reported	“5—totally agree” to a five-point Likert scale statement *“I have chronic pain.”*
Chronic stress	F1.F2	Questionnaire	self-reported	“5—totally agree” to a five-point Likert scale statement *“I repeatedly suffer from stress.”*
Regular smoking	F1,F2	Questionnaire	self-reported	“Yes” to *“Do you smoke regularly?”*
Daily alcohol	F1.F2	Questionnaire	self-reported	“5—totally agree or 4—partly agree” to a five-point Likert scale statement *“I use alcohol on a daily basis.”*

* B = baseline 2010–2011, F1 = Follow-up 2013–2014, F2, Follow-up 2016–2017. ** Each health risk factor was assigned a value of 1 if the definition was met, otherwise it was assigned a value of 0.

**Table 2 ijerph-17-09426-t002:** Health Risk Trends at Three Different Time Points.

	Baseline	Follow up 1	Follow up 2	Difference	Difference	Difference
	2010–2011	2013–2014	2016–2017	BL^®^ F1	F1^®^ F2	BL^®^ F2
**Participants**						
Total employees	651	530	523	−121	−7	−128
HRA participants	90.30%	81.30%	69.40%	−9.0	−11.9	−20.9
Percent males	83.50%	82.60%	80.80%	−0.9	−1.8	−2.7
Mean age (yrs)	43.8	45.1	44.2	1.3	−0.9	0.4
Percent ≤ 35 years	30.40%	25.80%	26.70%	−4.6	0.9	−3.7
Percent 36-45 years	17.50%	19.10%	25.30%	1.6	6.2	7.8
Percent 46-55 years	37.10%	34.20%	29.50%	−2.9	−4.7	−7.6
Percent ≥ 56 years	15.00%	20.90%	18.50%	5.9	−2.4	3.5
**Personnel Group**						
Executives	14.80%	22.70%	20.10%	7.9	−2.6	5.3
Local forest officers	46.80%	49.20%	55.40%	2.4	6.2	8.6
Officers	18.70%	14.20%	10.70%	−4.5	−3.5	−8.0
Terminal workers	12.10%	9.70%	13.80%	−2.4	4.1	1.7
Lumberjacks	5.80%	4.20%	-	−1.6	−4.2	−5.8
NA	1.90%	-	-	−1.9	-	−1.9
**Health risks**						
Physical inactivity	20.40%	15.70%	15.90%	−4.7	0.2	−4.5
Low cardiovascular fitness	8.10%	5.20%	6.40%	−2.9	1.2	−1.7
High blood pressure	39.90%	36.30%	31.10%	−3.6	−5.2	−8.8
High bmi	38.30%	33.00%	35.20%	−5.3	2.2	−3.1
Low vitality	1.60%	1.90%	2.50%	0.3	0.6	0.9
Musculoskeletal disorder	21.80%	26.80%	20.60%	5	−6.2	−1.2
Illness days due to MSD	7.20%	9.10%	7.20%	1.9	−1.9	0
Medication for cardiovascular disease	12.20%	14.60%	12.00%	2.4	−2.6	−0.2
Chronic pain		3.10%	3.40%		0.3	
Chronic stress		2.80%	5.10%		2.3	
Smoking		9.80%	9.30%		−0.5	
Alcohol		2.40%	1.10%		−1.3	
Total sum of (8) health risks	874	608	472	−266	−136	−402
Total sum of (12) health risks	-	685	592	-	−93	-
Avg of (8) health risks (SD)	1.49 ± 1.34	1.41 ± 1.31	1.30 ± 1.28	−0.08	−0.11	−0.19
Avg of (12) health risks (SD)	-	1.59 ± 1.48	1.30 ± 1.36	-	−0.29	-
**Overall risk levels**						
Low risk (0–1)	58.80%	54.50%	59.50%	−4.3	5	0.7
Moderate risk (2–3)	32.50%	34.60%	30.00%	2.1	−4.6	−2.5
High risk (4 or more)	8.70%	10.90%	10.50%	2.2	−0.4	1.8
Total	100.00%	100.00%	100.00%			

**Table 3 ijerph-17-09426-t003:** Health Risk Changes between 2010–2013, 2014–2017 and 2010–2017.

	**2010**		**2013**			
**PHASE 1 (*N* = 359)**	***n***	**%**	***n***	**%**	**difference**	***p***
Physical inactivity	62	17.3%	55	15.4%	−1.8%	0.494
Low cardiovascular fitness	22	6.3%	16	4.5%	−1.8%	0.327
High blood pressure	131	36.5%	124	34.5%	−1.9%	0.752
High bmi	121	33.7%	118	32.9%	−0.8%	0.749
Low vitality	4	1.1%	8	2.2%	1.10%	0.388
Musculoskeletal disorder	71	20.2%	103	28.8%	8.60%	0.001 **
Illness days due to MSD	27	7.6%	29	8.1%	0.50%	1
Medication for cardiovascular disease	39	10.9%	55	15.4%	4.50%	0.001 **
**PHASE 2 (*N* = 255)**	**2014**		**2017**			
Physical inactivity	37	14.6%	36	14.3%	−0.3%	1
Low cardiovascular fitness	12	4.7%	13	5.1%	0.40%	1
High blood pressure	84	32.9%	77	30.3%	−2.6%	0.401
High bmi	84	32.9%	90	35.3%	2.40%	0.327
Low vitality	3	1.2%	5	2.0%	0.80%	0.453
Musculoskeletal disorder	62	24.3%	52	21.1%	−3.3%	0.233
Illness days due to MSD	20	7.9%	17	6.8%	−1.1%	0.69
Medication for cardiovascular disease	23	9.1%	32	12.7%	3.70%	0.031 *
Chronic pain	5	2.0%	10	4.0%	2.00%	0.125
Chronic stress	10	4.0%	15	6.1%	2.10%	0.359
Smoking	23	9.1%	19	7.6%	−1.4%	0.424
Alcohol	3	1.2%	2	0.8%	−0.4 %	1
**TOTAL PERIOD (*N* = 253)**	**2010**		**2017**			
Physical inactivity	42	16.6%	36	14.4%	−2.2%	0.617
Low cardiovascular fitness	19	7.7%	12	4.8%	−2.9%	0.481
High blood pressure	92	36.4%	77	30.8%	−5.6%	0.237
High bmi	81	32.0%	93	36.9%	4.90%	0.023 *
Low vitality	1	0.4%	4	1.6%	1.20%	1
Musculoskeletal disorder	42	17.0%	40	16.1%	−0.9%	0.268
Illness days due to MSD	18	7.3%	19	7.6%	+0.3%	0.85
Medication for cardiovascular disease	21	8.3%	52	16.7%	8.40%	0.000 **

*p* values are based on the McNemar´s test for non-parametric paired nominal data. * *p* ≤ 0.05, ** *p* ≤ 0.001.

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
