# Peer review of "Eight-Year Health Risks Trend Analysis of a Comprehensive Workplace Health Promotion Program"

_ijerph, 2020, doi:10.3390/ijerph17249426_

Round 1

Reviewer 1 Report

General Comment:  Unique study looking at longitudinal health risks in a Finnish wood supply company.  This is important work that adds to the science on health risks over time in employees.  Even though the intervention is detailed elsewhere, more intervention description is needed.  It is unclear how the cohorts are different than the entire employee population that completed the HRA – did they actively participate in the WHP?  The design of the study does not support the assertion that the WHP slowed health risks – no control group or comparison group.

Abstract:   Include some indication of sample size.   

Background:  Line 34 – can you be more specific regarding the factors known to be associated with “poor employee health” (mental, cardiovascular, pain, etc.).  Line 59 – name/describe the categories.  Line 61- indicate the range of intervention periods.  Consider revision of purpose statement from research question to aim statements to match how you refer to these in the discussion section.     

Methods:  Even though intervention is described in detail elsewhere – a few sentences about the intervention would be helpful.  Also, description of the occupational health office available to employees is important especially since the role of the occ health office is addressed in the discussion.     

Annual total number of employees is noteworthy – is the turnover rate also know?  Important to know not only total number, but also the composition of those that remain….what percentage of employees are new year to year?  The turnover rate by year should be reported for better understanding on how the introduction of new employees may be affecting the trends over time.  Appendix 2 appears to show the variation of the employee population (same vs. new) but this could be made clearer in both the text and in the table.    A sentence describing the Polar BodyAge TM system would help the reader.  How many items are in the questionnaire for each category (demographics, physical activity, lifestyle change etc) and how long does it take to complete?  How many questions for the transtheoretical model, service provider’s own questions, and employers own questions?  Who collected biometric data?  What type of submax test was used for fitness (walk, step, etc).  Table 1 is great – helps reader understand how risk points assigned. Line 196 – does improved biometrics mean participants moved from abnormal (high risk = 1) to normal (no risk = 0) or any favorable change in biometric score?  What is the difference between the 12 health risks in this study and the 15 health risks used in comparison studies?

Results:  Successful vs. unsuccessful is unclear.  Who comprises the successful group?  It appears the unsuccessful is comprised of men and forest officers.  Is this a correct interpretation?  Table 2 reports HRA participants but this seems very different to the cohort sample sizes.  For example, at baseline 586 took the HRA (90.3% of 651).  Line 181 indicates cohort sample size is 359.  Might consider adding a line to Table 2 to report on those who participated in the intervention at each time point (359, 255).  Of employees taking the HRA there appears to be two groups – those who participated in the intervention and those who did not or do the cohorts simply reflect those with assessments at all time points? 

Discussion:   Line 328 – data supporting the “moderate level impact on biometrics” was not presented in the results section.

Conclusion: Line 419 – without a control group asserting that the WHP is what is affecting the employee health and slow health risk accumulation is a stretch.  It may be that being employed is what keeps health risks low because of the effect on mental health, financial status, access to health services, and/or daily physical activity.       

Reviewer 2 Report

Overall comment:

In this study the authors investigated the trends for both biometric data and self-reported health factors. To further develop the knowledge about effects from implementation-initiatives in worksite health promotion, further studies are of importance. This study contributes with new perspectives on the expectations for long term effects. Although the paper is interesting, I think it needs improvements. Comments for the authors are listed below.

Main comment:

One of the strengths in this paper is the long-term perspective on trends for health outcomes in employees within an organization. Although, I believe the manuscripts needs to be revised both in terms of clarity and language. There is also a need for improvements in the discussion section, by further deepen the discussion on trends for outcomes related to aging and possible effects generated from the interventions. The results could and should also be more throughly discussed in relation to other cohorts/long term interventions, to get a hint of a possible effect on slowing down the accumulation of health risks. The fact that some of the outcomes are dependent/related, are also of importance to discuss. More detailed comments are listed below.

Minor comments

Abstract:

1. I wonder whether it would be more appropriate to name the cohorts in a different manner. They are, from mine point of view, not three separate cohorts. This could be misinterpreted, especially when reading the abstract. Consider to rename the different time-periods of the study, ie. Phase 1, Phase 2, and Total period. This suggestion, goes through the whole manuscript. This is especially important as about half of the study sample is participating in all assessment periods.

Method:

2. P3, L 102-103. Inconsequense in writing Enso or ENSO in this section.

3. P3, L 110-111. You write, ”whereas the last four years concentrated more on the environmental context emphasizing workplace climate, stress management and mental resources ”. I find it hard, even when looking at the references, to understand the difference in focus for the last four years. It needs to be more clear what the similarities and differences for the two periods actually were. What policies and possibilities were available the whole study period, and what was only available the first/second period. Environmental context is physical environment in my point of view, relating to facilities and interior design. Consider to describe whether your mean the physical environment and facilities or more cultural aspects/psychosocial aspects of work envrionment. It is difficult to get a grip of what to expect from the last period, from the description in the manuscript. Please clarify. This section is the base for reccuring descriptions at page 9 and 11.

4. P3, L 112. The midpoint shift…. The description of the threee cohorts/phases are not yet described when this sentence arrives. This information becomes confusing.

5. P 3,L 132. It is described that the biometric assessments were executed with Polar Body AgeTM-system, but there is no information on the content of the biometrical outcomes until the end of the section, below the table. Consider to write these parts together, making it easier to get a grasp.

6. P 5, L 187. I do not find the figure appendix in the appendix intuitive and helpful. Consider to design the figure in another way. Flow chart?

7. P5, L 196. Different descriptions/wording are used for the biometric screenings. Could you harmonize?

8. P6, L228. The sentence “The mean age stayed almost the same…. What do you actually mean by “lost it share”? I don´t understand what this adds.

9. P7, Table 2. Consider to switch the order of the two last columns, so that Difference F1->F2, comes before Difference BL->F2, which might be more logic.

10. P7, Table 2. It would be of interest to know the standard deviation (the variation of the data) for the mean vales of the average number of health risks.

11. P7, L 242: Medication, not mediation?

Results:

12. P8, section 3.2. Again, consider the wording for cohort 1, 2 etc.

13. P 8, L 262-263. We know that different socio-economic status/educational level differ regarding health, and maybe also in the way interventions comes through. In your subgroup analysis, this is not considered. I believe this aspect should be mentioned in relations to the choice of analysis or interpretation of results. If the groups were to small, to perform analysis for SES, this should be mentioned.

14. P 8, L 273. You use the word successful or unsuccessful, which includes quite a strong “judging”. Maybe consider if the categorization could be described with participants with increased or decreased number of risk factors over time?

Discussion:

Overall comment: The five “observations” in the discussion section is, for me, not clearly linked to the most interesting or important findings, and must be improved. 

15. P 9, L 309. Is there something wrong with this sentence, I don´t understand.

16. L 316. First: You claim that the studied population had lower prevalence of health risks, compared to other studies. Please give examples from the referenced literature, and describe whether these studies are performed in groups that are comparable regarding i.e educational status/socioeconomical perspective.

17. L 328. Second: The number of employees with medication for CV disease increases over time, but there are no changes in Blood pressure outcomes. The fact that blood pressure is more common as we age should be discussed. Another aspect of this, is how the employees were taken care of when a High Blood Pressure were found during assessments. An increase of medication might be a result from actions taken based on the study measurements. This might also be a protective action for the individual in the long run. This should be discussed, together with a description (in the intervention section) about how different health assessments were taken care of, in terms of intervention activities.

18. P11, L336. Here you describe a methodological aspect. Maybe this section should be placed later in the discussion section, as it does not clearly refer to your main findings.

19. P11, L 344. As you conclude that the intervention program was slowed down, I believe you need to discuss the normality of declines related to age. You refer to literature, but actually do not compare your own results with others. 

20. The outcomes in the assessments are more or less associated/related. Consider to discuss this aspect in relation to your methodology.

Conclusions:

21. To be able to draw such a strong conclusion, there is a need to clearly compare and describe the current results, with other similar longitudinal cohorts/ intervention studies.

Round 2

Reviewer 2 Report

Overall comment:

The manuscript has been significantly improved from the first version. It is more clear and informative, and I believe the discussion is better balanced.

I still miss the description of the fitness test performed, and would like the authors to describe what type of test was performed and if availability, a reference to the validity of the test.

Otherwise, I mostly have comments on language and spelling. This needs to be thoroughly done all through the manuscript.

Some detailed comments are listed below.

Suggestion for methodological clarification:

P4, line 167: Polar own Fitness test for VO2max – is this test performed on a bicycle, test cycle? Please clarify what type of test is performed and if possible, the validity of the test.

Comments on language:

  1. P1, L 35. The first time BMI occurs, the abbreviation should be presented. Also, both “bmi and BMI” is used in different parts of the text.

2. P2, L 68 Byrne et al (2011) reported statistically significant improvements in low-risk proportion, physical inactivity, sufficient fruits and vegetables intake, smoking, seat belt usage and high stress during a seven-year incentive-based worksite wellness program [30].

This sentence is a little difficult to understand. Improvements in low-risk proportion? Do you mean improvements in the proportion of workers in low-risk regarding physical inactivity….?

  1. P 2, L 84, The purpose of this study was to examine long-term health risk trends in response to a comprehensive WHP program.

Even though you write the aim of the current study further down, it could be made more clear here, that the aim of the previous studies based on the WHP program has been to….

  1. P 8, L 284: reported stress stayed the same or at a higher level?

  1. P9, L 296 ”Likewise, a relatively large increase was found in self-reported musculoskeletal disorders during Phase 1” Instead of in the phase 1?

  1. P 9, Line 300: ”noted in four out of eight variables in the Phase 1 and six out of twelve in the Phase 2 and four out of ”.

The same goes here, it is languagewise not smooth to read ”the Phase 1”. Consider to rewrite to make the it more easy to read.

  1. P 9, L 304: ”These subgroup analyses were based on sex, age group and 305 personnel group comparisons”.

  1. P 10, L 318 – suggestion: ”among terminal workers and lumberjacks in all three periods, and among men in both Phase 1 and 2.

  1. P 10, L 323 each phase the proportion of low-risk participants varied? from 58.4 percent to 67.6 percent.

  1. P11, L 343 approaches i.e. nutrition and physical activity

  1. P 11 L 388-397. I highly appreciate the discussion section on blood pressure and ageing. There are some language check needed in this section.

  1. P 12, L 426. I do not understand this sentence. ”However, following investigations offer valuable insight of possible effects”. Do you mean, ”However, long-term studies evaluating WHP programs offer valuable insights of possible intervention effects”?

  1. P12, L 433: ” As an summation” might be replaced with ”To summarize, or To sum up?

  1. , L 498 “unsuccessful” lifestyle change groups in order to increase? the effectiveness of…

  1. P13, L 489 ”In summary, this study's results show that the accumulation of health risks as a function of aging was stopped”,

Was it stopped, or slowed down. Please harmonize with the conclusion in the abstract. Slowed down might be more realistic from my point of view.
